# Effect of Yeast Culture on Reproductive Performance, Gut Microbiota, and Milk Composition in Primiparous Sows

**DOI:** 10.3390/ani13182954

**Published:** 2023-09-18

**Authors:** Zhizhuo Ma, Ze Wu, Yu Wang, Qingwei Meng, Peng Chen, Jianping Li, Anshan Shan

**Affiliations:** 1College of Animal Science and Technology, Northeast Agricultural University, Harbin 150030, China; zhizhuoma@163.com (Z.M.); wuzeneau@163.com (Z.W.); wangyu205831@163.com (Y.W.); qingweimeng1990@163.com (Q.M.); 2Beijing Enhalor International Tech Co., Ltd., Beijing 100081, China; chenpengcau@gmail.com

**Keywords:** primiparous sows, prebiotics, reproductive efficiency, gut microbiota, milk

## Abstract

**Simple Summary:**

Compared with multiparous sows, primiparous sows have poor reproductive performance and insufficient nutritional reserves, resulting in a decreased growth performance in their offspring, which affects the efficiency of pig production. Yeast culture (YC) is rich in nutrients, and the supplementation of yeast culture in gestation and lactation diets is beneficial to improving the nutritional level of primiparous sows. Therefore, the objective of this experiment was mainly to investigate the effects of supplementing 0.5% and 0.8% yeast cultures into diets during gestation and lactation on the reproductive performance and intestinal health of primiparous sows, and the nutrient content of their colostrum and milk. The results showed that the dietary yeast cultures improved reproductive performance, improved the intestinal environment and energy metabolism, and increased the nutrient content of colostrum from primiparous sows. This study provides a theoretical basis for improving the performance of primiparous sows, and also provides a reference for the application of yeast cultures in sow production.

**Abstract:**

The objective of this study was to evaluate the effects of yeast culture (YC) on reproductive performance, gut microbiota, and milk composition in primiparous sows. A total of 60 primiparous sows were randomly assigned to the control group (CON) and YC group (0.5% YC during gestation and 0.8% YC during lactation) consisting of 30 replicates, with one sow in each. The results showed that dietary YC supplementation increased the piglet birth weight and backfat thickness at 28 d of lactation (*p* < 0.05). Dietary YC supplementation increased the apparent total tract digestibility (ATTD) of gross energy and calcium during lactation, the content of acetic acid and propionic acid at 110 d of gestation, and the content of acetic acid and butyric acid at 28 d of lactation in feces (*p* < 0.05). Furthermore, dietary YC supplementation decreased the abundance of Firmicutes, *Lachnospiraceae_XPB1014_group*, and *Terrisporobacter* (*p* < 0.05), and increased the abundance of *Prevotellaceae_NK3B31_group* and *Rikenellaceae_RC9_gut_group* (*p* < 0.05). Compared to the control group, dietary YC supplementation increased the fat and lactose content of the colostrum (*p* < 0.05). Metabolomics analysis showed that YC increased 26 different metabolites in the colostrum. Among them were mainly pantothenic acid, proline, isoleucine, phenylalanine, acylcarnitine, and other metabolites. In conclusion, these results suggested that dietary YC supplementation improves reproductive performance and gut health and increases the nutrient content in the colostrum of primiparous sows.

## 1. Introduction

The increasing demand for pig production requires modern sows to have a higher reproductive performance and nutritional status [1]. However, the widespread use of lean-type pig breeds has led to a negative nutritional balance in sows before and after production, affecting their health status. Compared to multiparous sows, primiparous sows have a lower feed intake during gestation and lactation, and need to mobilize more body reserves to feed piglets [2]. The nutritional intake of primiparous sows during lactation cannot meet the needs of lactation, growth, and development, which will lead to the body being in a catabolic state, and affect the reproductive life, and the growth performance of piglets [1,3]. Previous studies have shown that the loss of body reserves during lactation in primiparous sows affects reproductive performance and piglet production performance [4,5]. In addition, due to the high replacement rate of sows and the large proportion of primiparous sows, an insufficient nutritional intake of primiparous sows will cause significant economic losses to the pig industry [6]. Therefore, it is necessary to increase the nutritional intake of primiparous sows through nutritional regulation to improve the production performance of primiparous sows and the growth performance of the next generation of piglets.

Yeast culture (YC), as a prebiotic, consists of a small number of residual yeast cells, yeast metabolites, cell wall components, and part of the culture medium [7]. YC contains rich nutrients, including oligosaccharides, proteins, vitamins and other nutrients, as well as unknown “growth factors”, which are beneficial in improving the nutrition level and maintaining the health of animals [8]. In studies in ruminants, YC has been used to improve milk production in dairy cows [9], as well as beef tenderness [10]. With the continuous research on the function of YC, there is an increasing interest in the effects of adding YC to pig diets, especially for sows. Zhao et al. [11] found that the average daily feed intake (ADFI), the milk yield in the first week of lactation, and the dry matter content in the milk of lactating sows were significantly increased by adding YC to the diets of sows in late gestation and lactation. Kim et al. [3] also reported that the supplementation of YC into the diet during gestation and lactation improved the performance of sows. Although numerous studies have been conducted on the role of YC in ruminants and multiparous sows, there are few reports on the effects of YC on primiparous sows. Therefore, this experiment aimed to supplement YC into the diet of primiparous sows during gestation and lactation, to investigate the effects of YC on the reproductive performance, gut microbiota, and milk composition of primiparous sows.

## 2. Materials and Methods

### 2.1. Experiment Design

A total of 60 primiparous sows with a similar backfat thickness (14.26 ± 0.22 mm) at 30 d of gestation from Heilongjiang Great Northeast Animal Husbandry Co., Ltd. (Harbin, China) were used in this experiment. All sows were randomly divided into a control (CON) group (basal diet) and a YC group (basal diet with 0.5% YC during gestation and 0.8% YC during lactation) with 30 replicates in each. The YC is a fermentation product of *Saccharomyces cerevisiae* with a crude protein content ≥ 15% and a mannan content ≥ 1%. The basal diets (Table 1) were formulated to meet or exceed the nutritional requirements of gestating and lactating sows, as recommended by the National Research Council (NRC 2012) [12]. The diet was fed twice a day (8:00 a.m. and 2:00 p.m.) at 2.5 kg from day 30 to day 86 of gestation, and at 3 kg from day 86 to day 110 of gestation. All sows were reared in a single pen, and the temperature in the gestating house was controlled at 18 °C–20 °C. At 110 d of gestation, the sows were transferred to the farrowing room and started on a lactation diet, followed by daily feeding restriction until farrowing according to the sow’s actual condition. The birth weight, stillbirth, deformities, and mummification of piglets were recorded at the time of delivery. Within 24 h of birth, piglets within the same treatment group were cross-fostered, and litter sizes were adjusted to 9–12 piglets. Piglets were routinely treated (ear-notching, tail-docking, castration, and iron injection) within 3 days after farrowing. During this period, sows were fed 4 times a day (5:00 a.m., 10:00 a.m., 4:00 p.m., and 8:00 p.m.). All sows were fed 1 kg of the lactation diet on the day of farrowing, which was thereafter increased by 1 kg daily, until ad libitum feeding. In addition, sows and piglets could drink freely throughout the experimental period. The temperature in the delivery room was maintained between 20 °C and 23 °C. After 28 d weaning, the sows were transferred to the gestation room. Two estrus detections were performed in the presence of mature boars (9:00 a.m. and 4:00 p.m.), and the interval between weaning and estrus was calculated.

### 2.2. Sample Collection and Measurement

The backfat thickness at point P2 (6.5 cm from the midline over the last rib) of primiparous sows was measured via B-mode ultrasound (Renco Lean Meater type 7, Minneapolis, MN, USA) at day 29 of gestation, day 1 and day 28 of lactation, to calculate the backfat loss during lactation. The sows’ weights were measured on the day of farrowing and weaning, and the weight loss during lactation was calculated. In addition, the daily feed intake during lactation was recorded, and the ADFI was calculated. After production, the number born, live number, stillbirth number, deformity number, and weak litter number of the sows were calculated. The piglets were weighed for their birth weight and weaning weight. After farrowing, 6 sows in each group were selected via hand-milking to collect colostrum samples (20 mL) after the sows were injected with 1 mL of oxytocin. Part of the colostrum was stored at −20 °C for assaying the colostrum composition, and the other part was stored at −80 °C for metabolomics analysis. Likewise, a 20 mL milk sample per sow was collected on day 14 of lactation. All colostrum and milk samples were taken from the sow’s anterior, middle, and posterior teats and then pooled. Fecal samples were collected from 12 sows selected per treatment group from day 14 to 16 of lactation. The fecal samples from each sow on these days were mixed, and homogenized with 10 mL of 10% hydrochloric acid per 100 g fecal sample, and then stored at −20 °C until chemical analysis. Twelve sows per group were randomly selected at 110 d of gestation and 28 d of lactation (the day of weaning). Fresh fecal samples were collected and stored at −20 °C for subsequent short-chain fatty acid (SCFA) analysis. A portion of the fecal samples collected on day 28 of lactation were stored at −80 °C for subsequent microbial analysis.

### 2.3. Chemical Analysis of Feed and Feces

All fecal samples were thawed and dried at 60 °C to a constant weight. Feed and fecal samples were ground and passed through a 1 mm screen, and then analyzed for crude protein (method 984.13; AOAC, 2006) [13], crude fiber (Van Soest method) [14], calcium (method 935.13; AOAC, 2006), and phosphorus (method 946.06; AOAC, 2006). The gross energy was measured via a bomb calorimeter (Parr 6300 bomb calorimeter; Parr Instrument Company, Moline, IL, USA). The apparent total tract digestibility (ATTD, %) of nutrients was calculated according to the methods of Zhe et al. [15].

### 2.4. SCFA Analysis

Two grams of fecal sample was weighed into a 10 mL centrifuge tube, and 5 mL of deionized water was added. The samples were swirled for 30 s, left at 4 °C for 30 min, and then centrifuged at 10,000 rpm for 10 min. The supernatant was filtered through a 0.22 μm filter, and this was repeated three times. The final supernatant was added to 25% (*w*/*v*) metaphosphoric acid in a 5:1 ratio for SCFA assay via gas chromatography (Shimadzu GC-2010, Shimadzu, Kyoto, Japan) [16].

### 2.5. Microbial Analysis

The MO BIO-power stool DNA isolation kit was used to extract the total bacterial DNA from the fecal samples of the CON group and YC group (*n* = 12) sows at 28 d of lactation, according to the manufacturer’s protocols. Before sequencing, the integrity of the extracted genomic DNA was determined via 1% (*w*/*v*) agarose gel electrophoresis. The extracted fecal DNA samples were sent to Applied Protein Technology (Shanghai, China) for amplified pyrosequencing on the Illumina HiSeq PE250 platform. The V4 hypervariable region of the 16S rRNA gene was amplified with the primers 515F and 806R (5′-GTGCCAGCMGCCGCGGTAA-3′ and 5′-GGACTACHVGGGTWTCTAAT-3′, respectively).

The valid label was mapped to the OTU using UParse V7.0.1001, with a 97% similarity. Selected a representative sequence for each OTU. The ribosome database project (RDP) classifier version 2.2 was used to assign a classification level to each representative sequence. The relative abundance of each OTU was detected at different taxonomic levels.

### 2.6. Metabolite Extraction and Analysis

The collected colostrum and milk were thawed on ice, and extracted with 50% methanol buffer. Briefly, the sample was extracted in 20 μL using 120 μL of 50% methanol, vortexed for 1 min, and incubated for 10 min at room temperature. Extracts were stored overnight at −20 °C. After centrifugation at 4000× *g* for 20 min, the supernatant was transferred to a new 96-well plate. The samples were stored at −80 °C before LC-MS analysis. In addition, 10 μL of each extract was taken and mixed to prepare QC samples. All samples were collected via the LC-MS system, according to machine commands. The liquid phase conditions and mass spectrometer Q-Exactive Triple TOF 5600plus (SCIEX, Cheshire, UK) parameters were as described by Li et al. [17].

The XCMS software (Version 3.9.3) was used to pre-process the acquired mass spectrometry data. The LC-MS raw data were converted to the mzXML format using MSConvert software (Version 3.0.6150), and then processed through the XCMS, CAMERA (Version 1.43.2), and metaX (Version 1.4.19) toolboxes in the R software (Version 4.1.2). The combined retention time (RT) and *m*/*z* data were used to characterize each ion. Further screening of differential metabolites in the samples was conducted using metaX.

### 2.7. Statistical Analysis

One sow, due to miscarriage (YC group), was excluded from the experiment. For the analysis of reproductive performance, sows and their litters were used as the experimental units. For the analysis of milk composition, sows were used as the experimental unit. Data were analyzed via an independent samples *t*-test, using SPSS 25.0 software (SPSS Inc., Chicago, IL, USA). The test results were expressed as means ± standard deviation (SD), and *p* < 0.05 was considered to be significant.

The relative abundance data of the gut microbiota were analyzed using the GLIMMIX program of SPSS 25.0. The alpha diversity within the community was calculated, and the taxonomic community was evaluated, using QIIME software (Version 1.7.0). Student’s *t*-test was performed to calculate the differences in metabolite concentrations of the colostrum and milk, and the *p*-values for multiple tests were adjusted using FDR (Benjamini-Hochberg). To differentiate variables between groups, principal component analysis (PCA) and supervised partial least-squares-discriminant analysis (PLS-DA) were performed via metaX. We combined the conditions of ratio ≥ 2 or ratio ≤ 0.5, *p*-value < 0.05 and variable importance in the projection (VIP) ≥ 1.5 to screen for important features.

## 3. Results

### 3.1. Production Performance of Primiparous Sows

The results of the reproductive performance of primiparous sows are shown in Table 2. Dietary YC supplementation increased the piglet birth weight and backfat thickness at 28 d of lactation (*p* < 0.05). However, no significant difference was observed on the litter size at birth, number of piglets born alive, litter birth weight, stillborn piglets, number of low-body-weight piglets, number of weaned piglets, litter weaning weight, piglet weaning weight, ADFI during lactation, body weight at the beginning and end of lactation, loss of body weight during lactation, backfat thickness at the beginning of lactation, loss of backfat (BF) during lactation, and days to return to estrus (*p* > 0.05).

### 3.2. The ATTD of Primiparous Sows during Lactation

As shown in Table 3, dietary supplementation with YC increased the ATTD of gross energy and calcium in primiparous sows (*p* < 0.05). However, the ATTD of crude ash, crude fiber, crude protein, and phosphorus was not affected by dietary YC supplementation (*p* > 0.05).

### 3.3. The SCFA Concentration in Fecal Samples of Primiparous Sows

The SCFA concentrations in the feces are shown in Figure 1. Dietary YC supplementation increased the concentrations of acetic acid and propionic acid in the feces of 110-day-gestation sows, and increased the concentrations of acetic acid and butyric acid in the feces of 28-day-lactation sows (*p* < 0.05).

### 3.4. Fecal Microbiota Analysis of Primiparous Sows

The effect of dietary YC supplementation on the fecal microbiota of primiparous sows is shown in Figure 2. Compared to the CON group, dietary YC supplementation significantly decreased the good coverage index of fecal microbiota, and significantly increased the Shannon, Simpson, and Sobs indexes (*p* < 0.05) (Figure 2A–D). At the phylum level, the fecal microbiota of primiparous sows consisted mainly of phyla such as Firmicutes and Bacteroidetes (Figure 2E). However, dietary YC supplementation significantly decreased the abundance of Firmicutes (*p* < 0.05). At the genus level, the fecal microbiota of primiparous sows was mainly composed of the genera *Prevotellaceae_NK3B31_group*, *Ruminococcaceae_UCG-005*, and *Lachnospiraceae_XPB1014_group*. Among them, dietary YC supplementation significantly increased the abundance of *Prevotellaceae_NK3B31_group* and *Rikenellaceae_RC9_gut_group* (*p* < 0.05) and decreased the abundance of *Lachnospiraceae_XPB1014_group* and *Terrisporobacter* (*p* < 0.05).

### 3.5. Colostrum and Milk Composition

As shown in Table 4, dietary YC supplementation increased the fat and lactose content in the colostrum (*p* < 0.05).

### 3.6. Metabolomic Analysis of Colostrum

A total of 418 effective metabolites, mainly including lipids and lipid-like molecules, organic acids and derivatives, and organoheterocyclic compounds, were analyzed via LC-MS from the CON and YC groups (Appendix A). PCA and PLS-DA showed that the CON and YC groups were independently distributed, with no overlap (Figure 3A,B). The first principal component (PC1) explained 55.06% and 55.42% of the PCA and OPLS-DA, respectively. The second principal component (PC2) explained 13.03% and 12.73% of the PCA and OPLS-DA, respectively. This indicates that the metabolites in the CON and YC groups differed significantly with the supplementation of YC in the diets. In addition, the PLS-DA model parameters R^2^ = 0.7528 and Q^2^ = −0.8014 indicate that the PLS-DA model is valid and reliable (Appendix A).

The detected metabolites were further screened by combining the screening conditions of ratio ≥ 2 or ratio ≤ 0.5, *p*-value < 0.05 and VIP ≥ 1.5, and a total of 26 differential metabolites were significantly influenced by YC treatment (Appendix A). The differential metabolites mainly belonged to organic acids and derivatives, phenylpropanoids and polyketides, and organoheterocyclic compounds (Table 5). Among them, compared with the CON group, the YC group showed significantly decreased contents of styrene oxide, methionine, gerberinol, and heliannuol C, whereas the contents of 22 differential metabolites, including linsidomine cation, pilocarpine, and xanthine, were significantly increased (Figure 3C).

## 4. Discussion

In this study, the piglet weight at birth and the backfat thickness at 28 d of lactation of primiparous sows were improved via dietary supplementation with YC. The level of nutrition provided by sows to their offspring depends on the nutrition in their diet and the body’s nutritional reserves [18]. As we failed to observe an increase in ADFI in primiparous sows with YC supplementation, it is hypothesized that the increase in piglet weight at birth is attributed to the supplementation of YC in the diet to improve the nutritional reserves of primiparous sows. However, Kim et al. [3] reported that dietary YC supplementation did not influence the piglet birth weight. This may be due to Kim et al. supplementing 12 g/d of YC into sows’ gestation diets, whereas the present study supplemented 0.5% of YC into sows’ gestation diets. In addition, in the present study, on day 86 of gestation, sows were fed an increased amount (from 2.5 kg/d to 3 kg/d) due to fetal development, compared to 2 kg/d in Kim et al. Therefore, this may be one of the reasons for the inconsistent results of different studies. In addition, it has been shown that the supplementation of YC into the diet improves backfat thickness in lambs [19]. It is hypothesized that the increase in backfat thickness in different animals may be related to the nutrients, such as protein, vitamins, and oligosaccharides, in YC [8].

As an important indicator to measure the efficiency of digestion and absorption in animals, ATTD is important to animal health and development [20]. In the present study, we observed that dietary YC supplementation increased the ATTD of gross energy and calcium during lactation. Energy digestibility is the total energy digestibility of protein, carbohydrates, and lipids in the diet [21]. An increased energy digestibility contributes to offspring growth in primiparous sows [15]. In addition, calcium in the sow’s diet during gestation and lactation is also critical for offspring bone development [22]. Previous studies have shown that the addition of mannan, one of the yeast cell wall components, to the diet improves the gross energy digestibility [23,24]. In addition, it is reported that each kilogram of YC contains 1400 units of phytase [25]. This may be the reason why dietary YC supplementation improves the ATTD of calcium. However, Veum et al. [26] reported that dietary YC supplementation did not affect the apparent digestibility in sows. As there are few reports on the effect of YC on ATTD in sows, the exact reasons need to be further studied.

SCFAs are metabolites produced by gut microbes through the fermentation of fiber and resistant starch, and mainly include acetate, propionate, and butyrate [27]. In this study, dietary YC supplementation increased acetic acid and propionic acid concentrations at 110 d of gestation, as well as acetic acid and butyric acid concentrations at 28 d of lactation, in primiparous sows. It has been reported that SCFA is associated with maternal hepatic lipid metabolism during gestation [28]. Increased concentrations of acetic and propionic acids contribute to the regulation of lipid metabolism, anti-inflammatory, and feeding behaviors [29]. In addition, the SCFA has a direct relationship with the milk composition. By infusing different SCFAs into cows, Rook et al. [30] observed that acetic acid could increase the milk yield and the fat content in milk, and butyric acid could increase the fat content in milk. Therefore, it was hypothesized that supplementation with YC during gestation and lactation increased different SCFA concentrations in primiparous sows, which may be beneficial in improving lipid metabolism and nutrient composition in the milk of primiparous sows. In addition, a previous study suggested that one possible reason for the enhancement of volatile fatty acids by the *Saccharomyces cerevisiae* fermentation product was the increased lactic acid utilization, which improves the microbial environment, and increases the abundance of cellulolytic flora [31]. In this study, we observed that YC increased the abundance of *Prevotellaceae_NK3B31_group* by analyzing the gut microbiota and hypothesized that this could be one of the reasons for the increased SCFA concentration [32].

In this study, we observed a significant increase in Shannon, Simpson, and Sobs indexes in the YC group compared to the CON group. It is suggested that dietary YC supplementation helps to increase the richness and diversity of the gut microbiota, which improves the gut health of primiparous sows. As YC contains nutrients such as β-1, 3-glucan and mannan, this may account for the increased gut microbial richness and diversity [33,34]. In the mammalian gut, Firmicutes and Bacteroidetes are generally the two most abundant phyla [35]. Consistent with this finding, our study also observed that Firmicutes and Bacteroidetes are the most abundant phyla. In addition, by analyzing the microbial composition at the phylum level, we observed that dietary YC supplementation significantly decreased the abundance of Firmicutes. This is similar to the results of Liu et al. [36], who found that the abundance of Firmicutes decreased from 90.76% to 77.64% with the supplementation of YC in late-laying hens. As a dominant phylum in the gut, Firmicutes regulates host carbohydrate metabolism and lipid metabolism, and plays an important role in the maintenance of internal homeostasis [37]. The decrease in the abundance of Firmicutes in this study may imply that YC has a regulatory effect on glycolipid metabolism in lactating sows. It is hypothesized that YC may have expanded the competition among the gut microbiota, leading to a decrease in the abundance of *Lachnospiraceae_XPB1014_group*, belonging to the Firmicutes, and an increase in the abundance of *Rikenellaceae_RC9_gut_group* and *Prevotellaceae_NK3B31_group*, belonging to Bacteroides. Through further analyzing the species composition at the genus level, we found that dietary YC supplementation increased the abundance of *Rikenellaceae_RC9_gut_group* and *Prevotellaceae_NK3B31_group* and decreased the abundance of *Lachnospiraceae_XPB1014_group* and *Terrisporobacter*. *Rikenellaceae*, as a common bacterium in the gut, regulates nitrogen and protein metabolism, and promotes butyric acid synthesis, as well as negatively correlating with host obesity [38]. *Prevotella* is associated with a diet rich in carbohydrates and fiber, and makes a significant contribution to the production of SCFA (acetic, propionic, and butyric acids) [39]. *Lachnospiraceae* are involved in the host’s carbohydrate metabolism and are, likewise, one of the major SCFA-producing microbiotas [40]. In addition, by feeding silage to finishing pigs, Niu et al. [41] observed that *Terrisporobacter* and *Lachnospiraceae* were strongly associated with SCFA content. In the present study, all four of the genera that changed significantly with the supplementation of YC in the diet were correlated with SCFA concentrations, although the changes in the abundance of different microbes varied. It is hypothesized that YC may be involved in regulating energy metabolism in primiparous sows.

As the nutritional status of primiparous sows was improved by the supplementation of YC, we analyzed the components in the colostrum and milk. In this study, dietary supplementation with YC increased the fat and lactose contents in the colostrum compared to the CON group. A previous study showed that dietary YC supplementation increased the lactose content of colostrum from sows [11]. In addition, Dias et al. [42] observed that the supplementation of YC into the diet increased the fat and lactose content of milk from dairy cows. The uptake and oxidation of lactose and fat in colostrum provide energy for piglets and influence their survival and constant body temperature [43]. However, sow colostrum is characterized by a high immunoglobulin content and a low lactose and fat content [44]. In this study, dietary YC supplementation improved the low lactose and fat content of colostrum in sows. In addition, the yield and quality of milk affect the proper development of piglets, which, in turn, is related to the profitability of pork production [45]. Therefore, dietary YC supplementation resulted in a higher nutrient content in colostrum and milk from primiparous sows, which is significant for piglet development and pork production.

The quality of sow colostrum directly affects piglet mortality and growth performance [46]. We further performed a metabolomic analysis of the components in colostrum from primiparous sows. Dietary YC supplementation mainly increased organic acids and derivatives, and lipids and lipid-like molecules, mainly including pantothenic acid and acylcarnitine. Pantothenic acid, also known as vitamin B5, is phosphorylated by pantothenic kinase after entering the cell, and is converted to coenzyme A through a variety of intermediates in all tissues to function [47]. A previous study showed that piglets deficient in pantothenic acid developed severe diarrhea within 2–4 weeks, and uncoordinated locomotion at 4–7 weeks [48]. Additionally, the levels of pantothenic acid in sows’ blood and milk were influenced by the levels of pantothenic acid in the diet [46,49]. Therefore, dietary YC supplementation for primiparous sows is beneficial to piglet supplementation with pantothenic acid and improves piglet health. In addition, we observed that dietary YC supplementation to primiparous sows increased the levels of several amino acids, including proline, isoleucine, and phenylalanine. Proline is synthesized from glutamate in a series of steps, through enzymes such as γ-glutamyl phosphate and γ-glutamyl phosphate reductase, and is considered to be an important amino acid in the metabolic processes of cellular redox control, bioenergetics, and apoptosis and cancer [50,51]. As one of the branched-chain amino acids, isoleucine is involved in physiological functions such as protein metabolism, fatty acid metabolism, and glucose transport, and is conducive to promoting the body’s immune function [52]. Dietary phenylalanine enters the liver through the blood, and is eventually absorbed by the tissues to synthesize proteins, and the excess is hydroxylated by the liver enzyme phenylalanine-4-hydroxylase to form tyrosine, then hydroxyphenylpyruvate and, finally, metabolize to carbon dioxide [53]. In this study, dietary YC supplementation increased the levels of various amino acids, and provided more nutrients for the production of primiparous sows and the development of piglets. In addition, in the present study, we observed an up-regulation of acylcarnitine content. In the cytoplasm, fatty acids are converted to the form of acylcarnitines, which are transported by the acylcarnitine transporter into the mitochondrial matrix to generate energy through β-oxidation [54]. Increased acylcarnitine in the plasma of lactating sows has been reported to be beneficial for improving sow performance and the growth of weaned piglets [55]. Therefore, we hypothesized that dietary YC supplementation also promoted energy metabolism in primiparous sows, and that the increased acylcarnitine content of colostrum was beneficial in enhancing piglet growth.

There are several limitations to this study. First of all, the present study did not study piglets excessively, including their feed intake, detailed body weights at different stages, diarrhea, or gut microbiota. Additionally, considering that YC is thought to have an immunity-enhancing effect [11,56], we did not study the immunity indexes of primiparous sows or the immunity indexes of piglets. Therefore, we could not know whether the decrease in sow body weight loss and the increase in the weaning weight of piglets were related to the immunity-enhancing effect of YC. Additionally, whether the supplementation of YC had an effect on subsequent litter size in primiparous sows was limited by the experimental conditions and remained unobserved in this study. However, multiple beneficial effects of YC on the performance of primiparous sows were found in this study, and this study was conducted in a commercial swine farm setting, which is more in line with the actual production environment. Therefore, the supplementation of YC in the diets of primiparous sows is beneficial to the performance of sows and piglets, and to improvements in the efficiency of the pig farming industry.

## 5. Conclusions

This study suggested that dietary YC supplementation during gestation and lactation improved the reproductive performance of primiparous sows, increased energy metabolism as well as adjusted the microbial structure in the gut. In addition, dietary YC supplementation was able to increase a variety of nutrients in colostrum. Our results suggested that dietary YC supplementation is beneficial to the health of primiparous sows and the performance of their offspring and that YC is a suitable prebiotic for primiparous sows.

## Figures and Tables

**Figure 1 animals-13-02954-f001:**
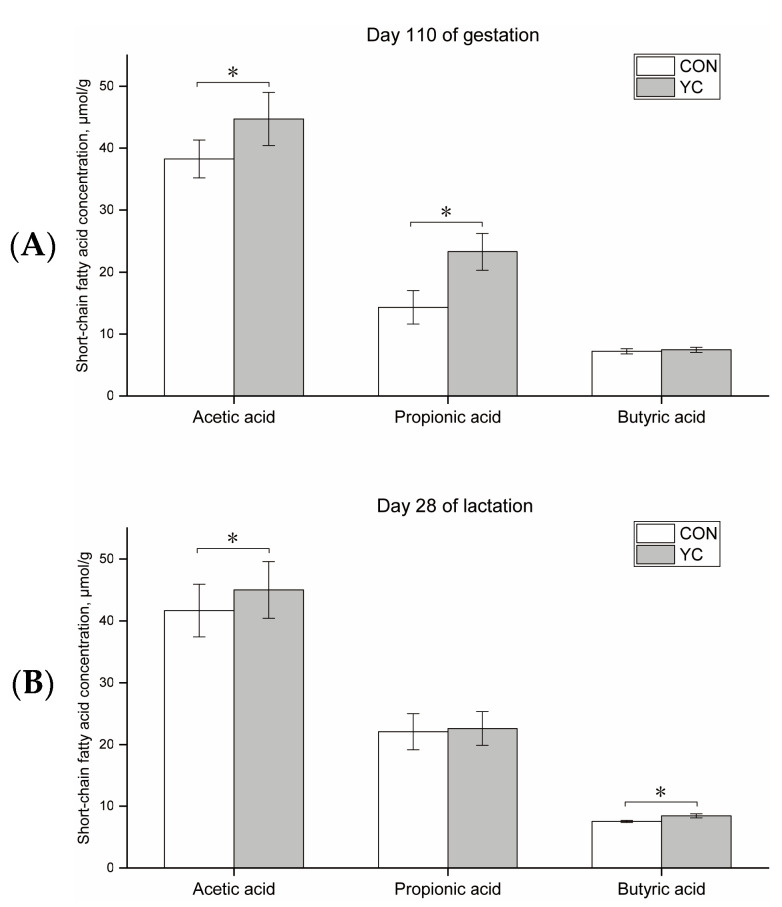
Short-chain fatty acid (SCFA) concentrations in the feces of primiparous sows during gestation and lactation. (**A**) SCFA concentration during gestation. (**B**) SCFA concentration during lactation. CON, control group; YC, yeast culture group. The data are expressed as mean ± standard deviation (*n* = 12). *, *p* < 0.05.

**Figure 2 animals-13-02954-f002:**
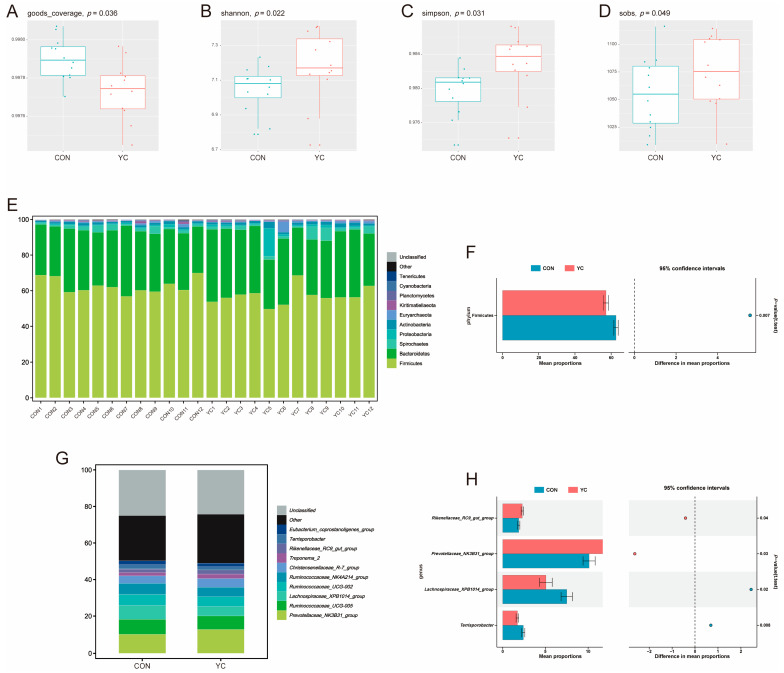
The microbiota analysis of feces during lactation in primiparous sows. (**A**–**D**) The good coverage index, Shannon index, Simpson index, and Sobs index. (**E**) The microbiota composition at the phylum level. (**F**) Differential microbiota at the phylum level. (**G**) The microbiota composition at the genus level. (**H**) Differential microbiota at the genus level. CON, control group; YC, yeast culture group. The data are expressed as mean ± standard deviation (*n* = 12).

**Figure 3 animals-13-02954-f003:**
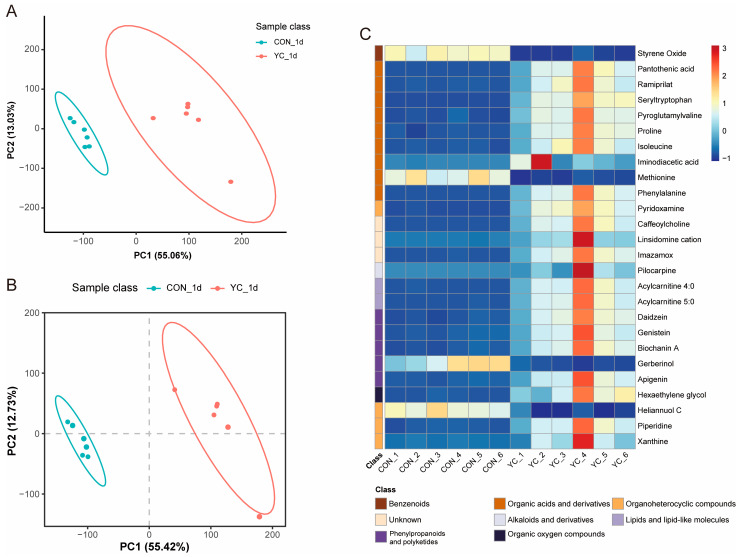
The metabolomic analysis of colostrum from primiparous sows. (**A**) Principal component analysis (PCA) of colostrum samples. (**B**) Partial least-squares-discriminant analysis (PLS-DA) of colostrum samples. (**C**) Heat map of differential metabolites. CON, control group; YC, yeast culture group, *n* = 6.

**Table 1 animals-13-02954-t001:** The composition and nutrient levels of basal diets (as-fed basis).

Items	Gestation	Lactation
Ingredients, %		
Corn	55.81	61.24
Wheat bran	9.23	5.00
Soybean meal	16.38	21.70
Expanded soybean	14.63	−
Fish meal	−	2.50
Soybean oil	−	5.00
L-lysine-HCL	0.04	0.27
DL-methionine	−	0.12
Chloride choline	0.15	0.15
NaCL	0.40	0.40
Calcium carbonate	0.16	0.17
Dicalcium phosphate	1.20	1.45
Vitamin and mineral premix ^1^	2.00	2.00
Total	100.00	100.00
Nutrient levels ^2^		
Net energy, MJ/kg	9.88	9.91
Crude protein	16.89	16.65
Calcium	0.86	0.92
Phosphorus	0.65	0.63
Lysine	0.73	1.11
Methionine	0.66	0.61

^1^ The vitamin and trace minerals supplied per kilogram of complete diet: 13,000 IU vitamin A; 3630 IU vitamin D_3_; 26.4 IU vitamin E; 240 mg vitamin K; 21.4 µg vitamin B_12_; 4.2 mg riboflavin; 12.5 mg d-pantothenic acid; 16.5 mg niacin; 20 mg Cu; 40 mg Mn; 160 mg Fe; 160 mg Zn; 0.4 mg I; 0.4 mg Cr; 0.4 mg Se. ^2^ The net energy, lysine, and methionine were calculated; the other values were measured.

**Table 2 animals-13-02954-t002:** The effect of dietary YC supplementation during gestation and lactation on the reproductive performance of primiparous sows ^1^.

Items	CON	YC	*p*-Value
Litter size at birth, heads	12.23 ± 2.16	12.45 ± 2.37	0.717
Number of piglets born alive, heads	11.93 ± 2.18	12.34 ± 2.33	0.487
Litter birth weight, kg	16.97 ± 3.29	18.47 ± 3.39	0.271
Piglet birth weight, kg	1.42 ± 0.04	1.49 ± 0.07	0.046
Stillborn piglets, heads	0.30 ± 0.53	0.10 ± 0.31	0.091
Piglets < 1.0 kg, heads	0.90 ± 0.96	0.52 ± 0.57	0.069
Number of weaned piglets, heads	11.03 ± 2.31	11.86 ± 2.13	0.158
Litter weaning weight, kg	87.18 ± 4.26	89.68 ± 6.58	0.648
Piglet weaning weight, kg	7.39 ± 0.19	7.53 ± 0.19	0.155
ADFI during lactation ^2^, kg/d	5.12 ± 0.23	5.10 ± 0.47	0.302
Body weight at 1 d of lactation, kg	138.4 ± 10.52	140.2 ± 9.44	0.776
Body weight at 28 d of lactation, kg	132.8 ± 9.08	134.8 ± 9.69	0.142
Loss of body weight during lactation, kg	6.30 ± 0.66	5.48 ± 0.60	0.755
Backfat thickness at 1 d of lactation, mm	18.12 ± 1.43	18.18 ± 1.22	0.395
Backfat thickness at 28 d of lactation, mm	17.84 ± 1.28	18.93 ± 1.04	0.031
Loss of BF during lactation ^3^, mm	0.30 ± 0.02	0.27 ± 0.03	0.255
Days to return to estrus, d	5.41 ± 0.91	4.90 ± 0.72	0.862

^1^ CON = control group; YC = yeast culture group. All values are expressed as means ± standard deviation (*n* = 30 in the control group and *n* = 29 in the YC group). ^2^ ADFI = Average daily feed intake. ^3^ BF = Backfat.

**Table 3 animals-13-02954-t003:** The effect of dietary YC supplementation on the nutrient apparent total tract digestibility in lactation primiparous sows ^1^.

Items (%)	CON	YC	*p*-Value
Crude ash	76.24 ± 0.44	75.22 ± 0.59	0.388
Crude fiber	13.36 ± 0.63	12.51 ± 0.47	0.361
Gross energy	83.10 ± 0.22	84.20 ± 0.26	0.049
Crude protein	81.60 ± 0.22	82.31 ± 0.42	0.302
Phosphorus	40.53 ± 0.44	40.84 ± 0.60	0.054
Calcium	70.12 ± 0.55	73.44 ± 0.72	0.032

^1^ CON = control group; YC = yeast culture group. All values are expressed as means ± standard deviation (*n* = 12).

**Table 4 animals-13-02954-t004:** The effect of YC on the colostrum and milk composition from primiparous sows ^1^.

Items	CON	YC	*p*-Value
Colostrum			
Fat, %	5.03 ± 0.16	5.19 ± 0.10	0.046
Protein, %	15.75 ± 0.33	15.92 ± 0.53	0.345
Lactose, %	3.47 ± 0.67	3.94 ± 0.77	0.027
Dry matter, %	28.04 ± 0.40	28.11 ± 0.77	0.804
Milk			
Fat, %	5.02 ± 0.11	5.12 ± 0.63	0.055
Protein, %	5.72 ± 0.37	5.81 ± 0.25	0.444
Lactose, %	5.21 ± 0.71	5.39 ± 0.11	0.060
Dry matter, %	19.67 ± 0.51	20.04 ± 0.73	0.163

^1^ CON = control group; YC = yeast culture group. All values are expressed as means ± standard deviation (*n* = 6).

**Table 5 animals-13-02954-t005:** Differential metabolites in colostrum from primiparous sows.

Metabolite	Class	Ratio ^1^	*p*-Value	VIP ^2^	MZ ^3^	RT ^4^
Styrene oxide	Benzenoids	0.05	7.70 × 10^−5^	1.96	121.08	1.41
Pantothenic acid	Organic acids and derivatives	60.50	1.03 × 10^−6^	2.33	220.13	3.18
Ramiprilat	Organic acids and derivatives	24.57	1.14 × 10^−6^	2.04	387.19	3.25
Seryltryptophan	Organic acids and derivatives	49.14	1.93 × 10^−7^	2.25	290.11	3.36
Pyroglutamylvaline	Organic acids and derivatives	14.82	6.94 × 10^−7^	1.90	227.10	3.34
Proline	Organic acids and derivatives	15.99	6.05 × 10^−3^	2.14	116.07	3.31
Isoleucine	Organic acids and derivatives	17.67	3.17 × 10^−6^	1.91	132.10	3.23
Iminodiacetic acid	Organic acids and derivatives	31.52	3.65 × 10^−3^	1.77	134.04	3.74
Methionine	Organic acids and derivatives	0.15	2.98 × 10^−7^	1.68	150.06	1.23
Phenylalanine	Organic acids and derivatives	8.55	4.39 × 10^−5^	1.69	166.09	3.54
Pyridoxamine	Organoheterocyclic compounds	6.63	2.36 × 10^−5^	1.61	169.10	3.15
Caffeoylcholine	Unknown	112.25	1.01 × 10^−8^	2.50	266.16	3.41
Linsidomine cation	Unknown	22.39	6.94 × 10^−4^	2.14	189.12	2.31
Imazamox	Unknown	57.85	4.80 × 10^−6^	2.32	304.15	3.19
Pilocarpine	Alkaloids and derivatives	102.54	1.52 × 10^−3^	2.34	226.15	1.86
Acylcarnitine 4:0	Lipids and lipid-like molecules	100.27	2.84 × 10^−8^	2.52	232.17	3.33
Acylcarnitine 5:0	Lipids and lipid-like molecules	76.79	2.16 × 10^−8^	2.46	246.19	3.45
Daidzein	Phenylpropanoids and polyketides	6.59	1.06 × 10^−5^	1.65	255.06	3.97
Genistein	Phenylpropanoids and polyketides	9.36	1.15 × 10^−5^	1.80	271.06	4.17
Biochanin A	Phenylpropanoids and polyketides	7.04	2.42 × 10^−5^	1.65	285.07	4.01
Gerberinol	Phenylpropanoids and polyketides	0.12	9.03 × 10^−6^	1.84	365.10	3.32
Apigenin	Phenylpropanoids and polyketides	9.24	3.32 × 10^−5^	1.78	269.04	4.18
Hexaethylene glycol	Organic oxygen compounds	11.15	1.42 × 10^−5^	1.56	300.19	3.31
Heliannuol C	Organoheterocyclic compounds	0.22	1.80 × 10^−4^	1.56	307.15	0.69
Piperidine	Organoheterocyclic compounds	9.14	2.04 × 10^−5^	1.61	86.10	1.76
Xanthine	Organoheterocyclic compounds	11.37	1.36 × 10^−3^	1.75	151.02	1.35

^1^ ratio = the average value of the YC group/average value of the CON group. ^2^ VIP = variable importance in the projection. ^3^ MZ = charge-to-mass ratio. ^4^ RT = retention time.

## Data Availability

The data presented in this study are available in the present article, and is shared with consent and in accordance with all authors.

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
