# Peer review of "Effect of Yeast Culture on Reproductive Performance, Gut Microbiota, and Milk Composition in Primiparous Sows"

_animals, 2023, doi:10.3390/ani13182954_

Round 1
Reviewer 1 Report
Comments on Animals-2585621
The manuscript entitled “Effect of yeast cultures on reproductive performance, gut microbiota and milk composition in primiparous sows” by Zhizhuo Ma et al. investigated the effects of dietary supplemented yeast culture (YC) on reproductive performance, gut microbiota, and milk composition by feeding trial with primiparous sows. The results showed that that dietary YC supplementation improves reproductive performance, gut health, and increases nutrient content in the colostrum of primiparous sows. The experiment is fit for the scope of Animals. The content of this manuscript was important for the sow production. The writing of the manuscript was very good. But I think the current manuscript needs to be clearly revised due to the following concerns.
Abstract: According to the “Instructions to Authors”, abbreviations should be defined the first time they appear in abstract. “CON” in Line 23 should be added in parentheses after the written-out form.
Line 78: “castration, iron injection” → “castration and iron injection”.
Line 126: Please check the method number of calcium. “958.01” is “Phosphorus (Total) in Fertilizers Spectrophotometric Molybdovana”.
Line 211and Line 241: If the results are not significant, there is no need to mention them.
L351: The dietary supplementation of YC increased the fat. Discuss more on it.
Line 464: “and -mannanase supplementation” → “and β-mannanase supplementation”
Table 1: It is better to add net energy or metabolism energy level in Table 1 and 2 because the two systems are more comprehensive in energy partitioning. Also, the CP content of CON lactation diet is low. Could you please explain in more detail.
Table 2: Add more data in sow performances based on 2.2. Sample Collection and Measurement, like body weight, backfat thickness, number of piglets weaned, etc.
Figure 1: Please check the details including the black dot in the bottom left corner and the frame line of “CON” “YC” in the upper right corner.
Reviewer 2 Report
I found the research article titled 'Effect of yeast cultures on reproductive performance, gut microbiota and milk composition in primiparous sows' to be captivating, and I engaged with the manuscript with a high level of interest. The paper’s alignment with the scope of the journal is commendable. However, I believe that some improvements are necessary to enhance its quality:
- I recommend revising the simple summary. According to the guidelines outlined for authors in the journal's instructions (https://www.mdpi.com/journal/animals/instructions), the simple summary should effectively convey the research problem under scrutiny, the study's precise goals and objectives, pivotal and pertinent findings, the conclusions drawn from the study, and their potential societal implications. Additionally, it is important that the summary is adapted to a general audience, devoid of intricate technical language.
- Additionally, it is advisable to introduce keywords that are distinct from those already present in the title. For instance, consider integrating terms such as "prebiotics," "feed additives," and "reproductive efficiency." Of course, if the authors can propose more fitting keywords, I encourage their inclusion.
- I recommend the inclusion of specific details within the Materials and Methods section, if feasible. Providing the average weight of the sows involved in the study would be valuable. Furthermore, detailing the characteristics of the prebiotic employed could contribute to the comprehensiveness of the section.
- To enhance the quality of the discussion, I propose an expansion of this section. Incorporating a discussion of the study’s limitations would offer a more balanced perspective. Additionally, addressing the practical implications of the findings would enrich the discussion and provide readers with insights into the potential practical applications of the study.
Specific comments:
L 80: It would be advisable to include the citation number for NRC (2012) as well.
Table 1: To improve readability, consider adding borders to the nutrient level row.
L 125: Also, including the citation number for AOAC (2006) would be beneficial.
Table 5: I would recommend enhancing the readability of this table. Additionally, I suggest that the authors include all the abbreviations used as footnotes, even if they have already been mentioned in the text.
LL 280-282: I suggest including in the text the difference in inclusion levels between the two research studies.
Reviewer 3 Report
The presented manuscript contains the results of interesting research on the reproductive performance of primiparous sow. The Authors showed that supplementation of YC in gestation and lactation diets had a beneficial effect on:
- body weight of piglets at birth;
- intestinal health of sow by improving the diversity of intestinal microflora in females;
- fat and lactose content in colostrum and lactose in milk;
- piglet health (increase in the content of pantothenic acid in colostrum).
Notes to the text:
1. What yeast cultures were used in the study?
2. Were the sows of a similar age? Did they differ in body weight?
3. Feeding pregnant sows is 2.5 kg of feed. Was this amount administered throughout the pregnancy? There was feeding sows differentiated due to the development of fetuses after the 90th day of gestation?
4. Table 1: NaCL instead of Nacl.
5. There is no information about the collection of colostrum and milk samples: were they the same teats? Was milk collected once or more times during lactation – on which day of lactation? Was there an injection of oxytocin?
6. The study did not report the number of weaned piglets. Were there losses in piglet rearing from birth to the day of weaning – the Authors of these data do not provide. If there were losses of piglets, what was their cause?
7. Table 2: Litter size at birth, heads as well as number of piglets born alive, heads.
Round 2
Reviewer 2 Report
Good job!